# HIV, asymptomatic STI, and the rectal mucosal immune environment among young men who have sex with men

Vanessa E. Van Doren [1] *, S. Abigail Smith[1], Yi-Juan Hu[2], Gregory Tharp[3], Steven Bosinger[3,4], Cassie G. Ackerley[1], Phillip M. Murray[1], Rama R. Amara[3,4], Praveen K. Amancha[1], Robert A. Arthur[5], H. Richard Johnston[5], Colleen F. Kelley[1,6]

1 The Hope Clinic of the Emory Vaccine Center, Division of Infectious Diseases, Department of Medicine, Emory University School of Medicine, Atlanta, Georgia, United States of America, 2 Department of Biostatistics and Bioinformatics, Rollins School of Public Health, Emory University, Atlanta, Georgia, United States of America, 3 Emory National Primate Research Center, Emory University, Atlanta, Georgia, United States of America, 4 Department of Microbiology and Immunology, Emory University, Atlanta, Georgia, United States of America, 5 Emory Integrated Computational Core, Emory University, Atlanta, Georgia, United States of America, 6 Grady Health System, Atlanta, Georgia, United States of America

* vvandor@emory.edu

**Data Availability Statement:** All relevant data are within the paper and its Supporting information files.

## Abstract

Young men who have sex with men (YMSM) are disproportionately affected by HIV and bacterial sexually transmitted infections (STI) including gonorrhea, chlamydia, and syphilis; yet research into the immunologic effects of these infections is typically pursued in siloes. Here, we employed a syndemic approach to understand potential interactions of these infections on the rectal mucosal immune environment among YMSM. We enrolled YMSM aged 18–29 years with and without HIV and/or asymptomatic bacterial STI and collected blood, rectal secretions, and rectal tissue biopsies. YMSM with HIV were on suppressive antiretroviral therapy (ART) with preserved blood CD4 cell counts. We defined 7 innate and 19 adaptive immune cell subsets by flow cytometry, the rectal mucosal transcriptome by RNAseq, and the rectal mucosal microbiome by 16S rRNA sequencing and examined the effects of HIV and STI and their interactions. We measured tissue HIV RNA viral loads among YMSM with HIV and HIV replication in rectal explant challenge experiments among YMSM without HIV. HIV, but not asymptomatic STI, was associated with profound alterations in the cellular composition of the rectal mucosa. We did not detect a difference in the microbiome composition associated with HIV, but asymptomatic bacterial STI was associated with a higher probability of presence of potentially pathogenic taxa. When examining the rectal mucosal transcriptome, there was evidence of statistical interaction; asymptomatic bacterial STI was associated with upregulation of numerous inflammatory genes and enrichment for immune response pathways among YMSM with HIV, but not YMSM without HIV. Asymptomatic bacterial STI was not associated with differences in tissue HIV RNA viral loads or in HIV replication in explant challenge experiments. Our results suggest that asymptomatic bacterial STI may contribute to inflammation particularly among YMSM with HIV, and that future research should examine potential harms and interventions to reduce the health impact of these syndemic infections.

**Funding:** This work was supported by the National Institutes of Health: R01AI128799 (CFK), R01HD092033 (CFK), P30AI050409, P51OD011132. The funders had no role in study design, data collection and analysis, decision to publish, or preparation of the manuscript.

**Competing interests:** The authors have declared that no competing interests exist.

## Author summary

Young men who have sex with men (YMSM) are disproportionately affected by HIV and asymptomatic bacterial sexually transmitted infections (STI) including gonorrhea, chlamydia, and syphilis. However, the health effects of these infections are not typically studied together. In this study, we enrolled YMSM ages 18–29 with and without HIV and/or asymptomatic bacterial STI to study the immunologic effects of these infections, and their interactions, on the rectal mucosa. We found that HIV was associated with differences in the cellular make-up of the rectal tissues, and that STI was associated with an increase in the detection of potentially dangerous bacteria in the rectum. When we examined tissue gene expression, we found that STI was associated with inflammation only among YMSM with HIV, but not those without HIV. We did not see an effect of STI on differences in tissue viral loads among YMSM with HIV or in HIV replication in rectal explant experiments in YMSM without HIV. Our results suggest that asymptomatic bacterial STI may contribute to inflammation particularly among YMSM with HIV, and that future research should examine potential harms and interventions to reduce the health impact of these syndemic infections.

## Introduction

HIV and bacterial sexually transmitted infections (STI), including gonorrhea, chlamydia, and syphilis, are disproportionately concentrated among young men who have sex with men (YMSM), particularly racial and ethnic minority YMSM attributed to the pervasive social inequities and discrimination facing this population [1]. In 2019 in the United States, men who have sex with men (MSM) accounted for 69% of new HIV diagnoses, and 21% of these were among young people aged 13–24 [2]. The burden of bacterial STI is also profound among YMSM. People aged 15–24 years make up a quarter of the United States population but accounted for 61% of chlamydial infections in 2020 [3] and nearly half of new STIs every year [4]. In the United States, syphilis cases have increased 52% since 2016 with MSM and racial/ethnic minorities again disproportionately impacted [3]. However, research and interventions to control and improve outcomes for HIV and STI are commonly siloed as separate entities, which can hinder progress. Syndemic theory provides a framework to study co-occurring infections (e.g., HIV and bacterial STI) to better understand the pathways through which they interact biologically and potentially increase their negative overall health burden on a population [5,6]. Therefore, a more holistic approach to improving health for YMSM will consider the impact and interactions of these infections.

The gut mucosa is a critical site for both HIV transmission and immunopathogenesis. Approximately 70% of HIV infections in MSM are thought to occur via exposure during receptive anal intercourse (RAI), which has an 18-fold and 50-fold higher HIV transmission probability per exposure event when compared to vaginal or penile exposure, respectively [7]. While effective antiretroviral therapy (ART) has resulted in remarkable improvements in morbidity and mortality, HIV infection results in profound, chronic perturbations to the immune system that are particularly evident in the gut [8]. Similarly, the gut, and rectum in particular, is an important site for acquisition of bacterial STI among MSM. Approximately 85% of bacterial STIs in the rectum are asymptomatic, underpinning the CDC recommendation to screen for asymptomatic bacterial STI in sexually active MSM with multiple partners every 3 months [9]. Among MSM without HIV, bacterial STI has been associated with increased acquisition of

HIV, thought to be mediated through inflammation including compromised mucosal integrity, local recruitment of HIV target cells, and disruption of the local cytokine milieu [7,10]. Among MSM with HIV, the health effects of asymptomatic bacterial STI are less studied but could result in amplification of chronic inflammation through similar mechanisms and/or increases in tissue HIV replication, further entrenching the HIV reservoir in the critical gut compartment [11].

Because of their elevated risk of HIV acquisition, YMSM also face an increased burden of associated chronic inflammation over their lifetime [12]. They also face an elevated risk of bacterial STI acquisition. The inflammatory impact of bacterial STI, and whether this is exacerbated in the setting of concomitant HIV, is not yet fully elucidated. Given the syndemic burden of both HIV and asymptomatic bacterial STI among YMSM, we undertook this study to better understand the immunologic effects of these infections and their interaction on the rectal mucosal immune environment including the cellular composition, transcriptome, and microbiome. We chose to focus on four populations of YMSM: those without HIV, those with HIV on ART, and those with and without asymptomatic bacterial STI to evaluate both the impact of each of these diseases in isolation as well as their potential multiplicative effects. We hypothesized that asymptomatic bacterial STI would be associated with rectal mucosal inflammation among YMSM without HIV and that the baseline immunologic perturbations in the rectal mucosa of YMSM with HIV would be augmented by the inflammatory effects of asymptomatic bacterial STI. A comprehensive characterization of the immunologic effects of both HIV and asymptomatic bacterial STI on the rectal mucosa of YMSM will advance our understanding of the syndemic health burden of these infections and could illuminate pathways for new interventions to reduce their impact on this marginalized and underserved population.

## Results

### The clinical cohort

A total of 105 YMSM were enrolled into a cross-sectional study with peripheral blood and rectal mucosal sampling including collection of swabs for secretions and pinch biopsies, both via rigid sigmoidoscopy. We enrolled 14 YMSM with HIV and asymptomatic bacterial STI, 15 YMSM with HIV and without asymptomatic bacterial STI, 28 YMSM without HIV and with asymptomatic bacterial STI, and 48 YMSM without HIV and without asymptomatic bacterial STI. Demographic and clinical characteristics are presented in Table 1. All participants were aged 18–29 years and in good health. Participants with asymptomatic STI were diagnosed with rectal gonorrhea (GC), chlamydia (CT), and/or latent syphilis (TP) prior to mucosal sampling, and appropriate antibiotic treatment was provided immediately after study sampling. Participants with HIV were well-controlled on ART (median CD4 = 635 cells/uL, median viral load (VL)<20 copies/mL), and a minority of participants without HIV were taking oral pre-exposure prophylaxis (PrEP). The majority of participants across groups had positive serologies for cytomegalovirus (CMV). HSV-2 was more prevalent among YMSM with HIV, and there was no difference in HSV-1 positivity between groups. Participants reported similar numbers of receptive anal intercourse partners in the last 12 months across groups.

### The cellular composition of the rectal mucosa among YMSM with HIV demonstrated profound differences as compared to YMSM without HIV, but with little impact of asymptomatic STI

A total of 23 immune cell subsets were characterized by flow cytometry in blood and 26 subsets were characterized with rectal mucosal tissue samples as described in S1 Table, including

**Table 1. Demographic and clinical characteristics of the participants included in the study.**

| Characteristic | HIV$_{pos}$STI$_{pos}$ n = 14 | HIV$_{pos}$STI$_{neg}$ n = 15 | HIV$_{neg}$STI$_{pos}$ n = 28 | HIV$_{neg}$STI$_{neg}$ n = 48 | p-value |
|---|---|---|---|---|---|
| Median age in years (range) | 22 (18–29) | 22 (19–26) | 21 (18–28) | 21 (18–24) | 0.04* |
| Race n (%) | | | | | 0.3 |
| White | 0 (0%) | 2 (13.3%) | 1 (3.6%) | 4 (8.3%) | |
| Black | 14 (100%) | 12 (80%) | 26 (92.9%) | 44 (91.7%) | |
| Other | 0 (0%) | 1 (6.7%) | 1 (3.7%) | 0 (0%) | |
| CD4 (cells/uL): median (range) | 608 (232–1706) | 657 (403–1370) | N/A | N/A | 0.3 |
| Plasma viral load (copies/mL): median (range) | <20 (<20–279) | <20 (<20–95) | N/A | N/A | 0.4 |
| HSV-1+ n (%) | 7 (50%) | 12 (80%) | 18 (64.3%) | 22 (45.8%) | 0.09 |
| HSV-2+ n (%) | 10 (71.4%) | 11 (78.6%) | 6 (21.4%) | 6 (12.5%) | <0.01* |
| CMV+ n (%) | 14 (100%) | 15 (100%) | 26 (92.9%) | 43 (89.6%) | 0.5 |
| GC+ n (%) | 4 (28.6%) | N/A | 8 (29%) | N/A | 1 |
| CT+ n (%) | 4 (28.6%) | N/A | 10 (35.7%) | N/A | 0.6 |
| TP+ n (%) | 1 (7.1%) | N/A | 2 (7%) | N/A | 1 |
| Multiple STI+ n (%) | 5 (35.7%) | N/A | 8 (29%) | N/A | 0.6 |
| # of partners in past 12 months n (%) | | | | | 0.6 |
| 0–5 | 9 (64.3%) | 7 (46.7%) | 19 (67.9%) | 31 (64.6%) | |
| 6–20 | 5 (34.7%) | 7 (46.7%) | 7 (25%) | 16 (33.3%) | |
| 21+ | 0 (0%) | 1 (6.7%) | 2 (7.1%) | 1 (2.1%) | |
| On PrEP n (%) | N/A | N/A | 4 (14.3%) | 2 (4.2%) | 0.1 |

HIVpos = YMSM with HIV; HIV$_{neg}$ = YMSM without HIV; STI$_{pos}$ = YMSM with STI; STI$_{neg}$ = YMSM without STI; HSV-1 = herpes simplex virus 1; HSV-2 = herpes simplex virus 2; CMV = cytomegalovirus; GC = *Neisseria gonorrhea*; CT = *Chlamydia trachomatis*; TP = *Treponema pallidum*/syphilis; PrEP = pre-exposure prophylaxis; * = p < 0.05.

innate [neutrophils, monocyte/macrophages, myeloid and plasmacytoid dendritic cells, Natural Killer cells (NK), γδ T cells, and mucosal-associated invariant T cells (MAIT)] and adaptive [B cells, CD4+ and CD8+ T cells] immune cell subsets. T cells were further characterized based on naïve vs memory phenotype (CD45RA, CCR7), tissue residence (CD69$^{+}$CD103$^{+/-}$), HIV susceptibility marker expression (CCR5, a4ß7, Ki67), regulatory phenotype (FOXP3$^{+}$CD25$^{+}$), and cytokine production upon mitogen stimulation (IL17A, IFNγ, TNFα).

Given our modest sample size and multiple parameter outcomes of interest, we first used discovery-based linear decomposition modeling (LDM), which is a single analysis pathway that combines both global tests of effect as well as tests of individual outcomes and allows for inclusion of all four study groups into a single model that controls for multiple comparisons [13]. LDM also enables the inclusion of interaction terms to assess for effect modification. In this case, we were interested in interaction between HIV and STI, meaning that the effect of STI on immune cell subsets could differ based on the participant's HIV status. If significant interaction is identified, stratified analyses are indicated to determine the nature of the interaction. In this case, significant interaction between HIV and STI was not present in either the blood (p = 0.8) or rectal mucosal (p = 0.1) cell subset analyses, and stratified analyses were not pursued.

Because there was no interaction detected, we next conducted LDM global tests of significance with all four study groups included in the model. The LDM global test of significance evaluating the effect of asymptomatic STI, controlling for HIV status, was significant (p = 0.02) for blood cell subsets and marginally significant (p = 0.06) for rectal mucosal cell subsets. In contrast, the LDM global test of significance evaluating the effect of HIV was highly significant for both blood (p = 0.001) and rectal mucosa (p = 0.0002).

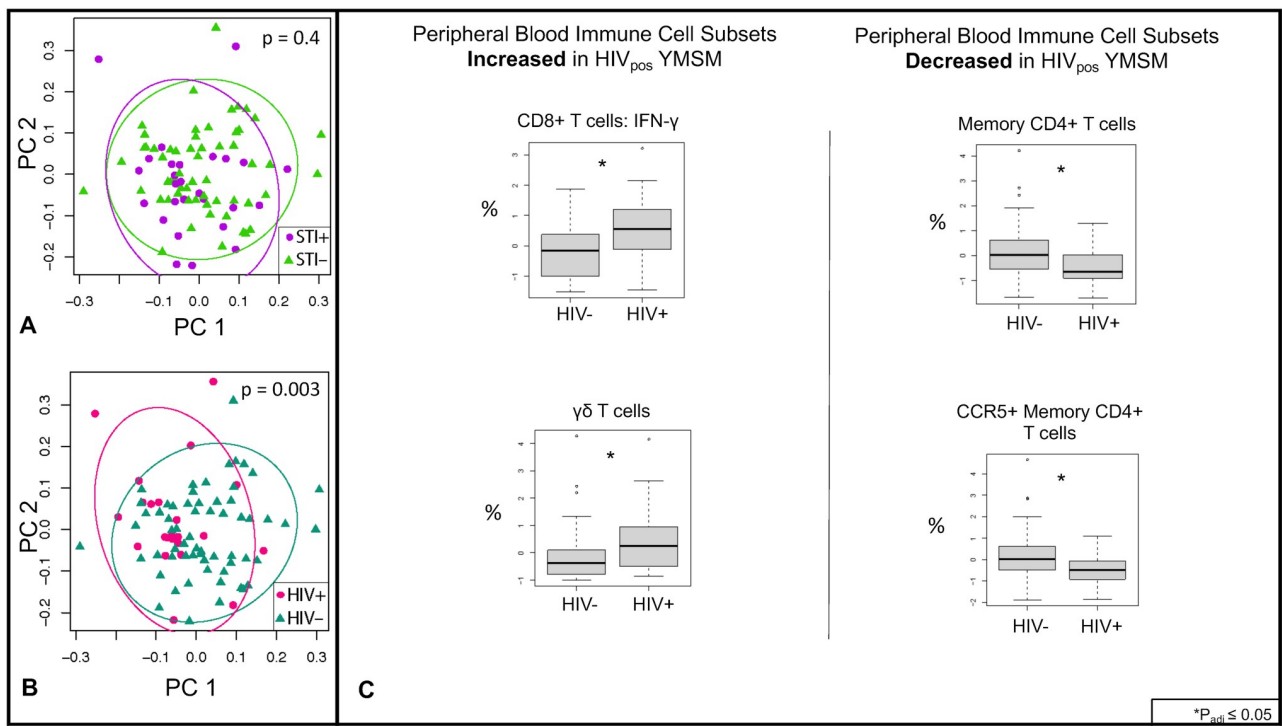

**Fig 1. Peripheral blood immune cell subsets in YMSM with and without HIV and asymptomatic rectal STI.** Twenty-three innate and adaptive cell subsets from peripheral blood were analyzed by flow cytometry. (A) There was no significant separation between YMSM with and without asymptomatic STI as analyzed by PERMANOVA. Purple dots represent STI-positive YMSM; green triangles represent STI-negative YMSM. (B) Significant separation was seen between YMSM with and without HIV. Pink dots represent HIV-positive YMSM; teal triangles represent HIV-negative YMSM. (C) After adjusting for STI, linear decomposition modeling (LDM) identified four cell subsets that were significantly (FDR<5%) different between YMSM with and without HIV. Cell subset percentages were log transformed for analyses. $HIV_{pos}$ = YMSM with HIV; $HIV_{neg}$ = YMSM without HIV.

Two-dimensional principal coordinates analyses (PCoAs) were constructed to better visualize global differences in cell subsets between YMSM with and without HIV. Blood data are presented in Fig 1, and rectal mucosal data are presented in Fig 2. Similar to the LDM results, significant separation was seen between YMSM with and without HIV in both blood (Fig 1B; p = 0.003 by PERMANOVA) and rectal mucosa (Fig 2B; p = 0.0002 by PERMANOVA), controlling for STI status. There was no significant separation by STI status, controlling for HIV (Fig 1A; p = 0.4 for blood and Fig 2A; p = 0.1 for rectal mucosa by PERMANOVA). Given the minimal effect detected for asymptomatic STI on global cellular subsets in blood and rectal mucosa, we concentrated further analyses on the effect of HIV.

Next, using LDM to assess differences between groups in individual cell subsets, we identified four subsets that were significantly different in blood between YMSM with and without HIV, including an increased percentage of CD8+ IFNγ T cells and γδ T cells and a decreased proportion of CD4+ memory and CD4+ memory CCR5+ T cells among YMSM with HIV (Fig 1C; $p_{adj}$<0.05). We also noted expected differences in other cell subsets such as a decrease in the percentage of total CD4+ T cells and increased percentage of total CD8+ T cells in the blood of YMSM with HIV, but these did not meet our stringent adjusted p-value cut-off. In contrast, there were 17 individual cell subsets that were significantly different in the rectal mucosa between YMSM with and without HIV (Fig 2C; $p_{adj}$<0.05). Among CD8+ T cell subsets, the proportion of total, memory, non-tissue-resident memory (TRM), IFNγ+, TNFα+, and Ki67 + cell subsets were significantly increased in the rectal mucosa of YMSM with HIV. In

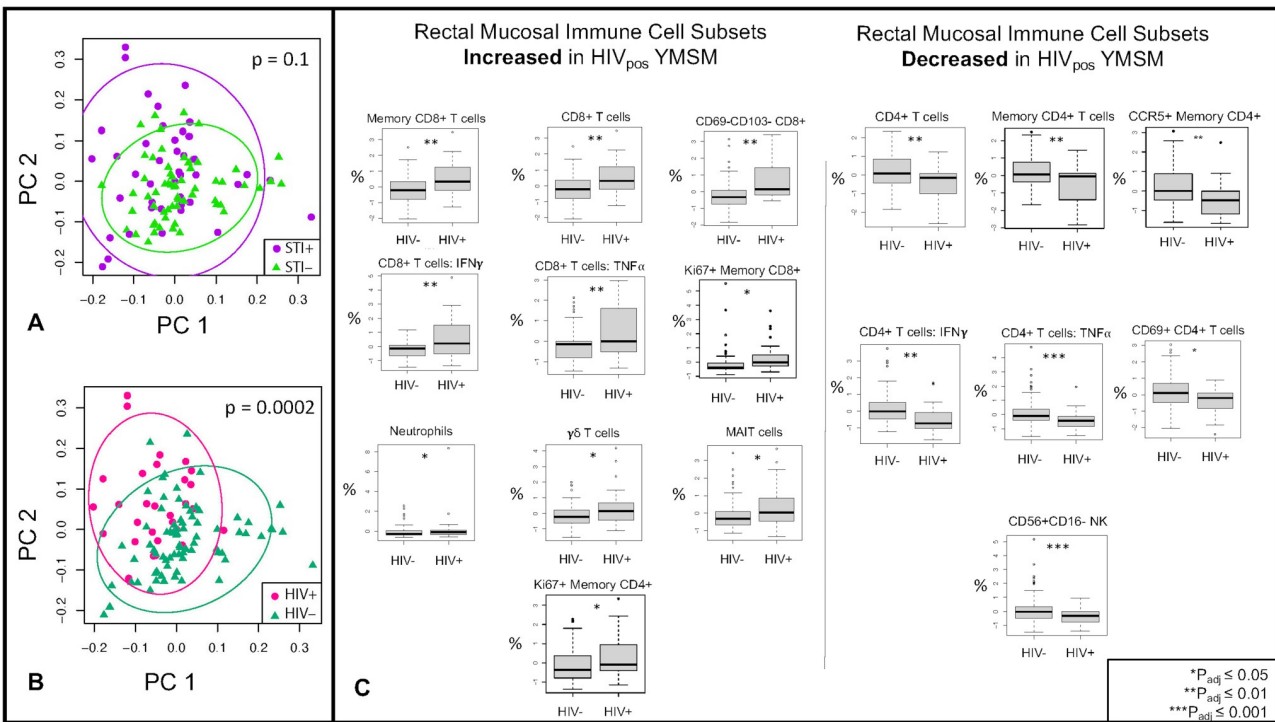

**Fig 2. Rectal mucosal immune cell subsets in YMSM with and without HIV and asymptomatic rectal STI.** Twenty-six innate and adaptive cell subsets from rectal mucosal tissues were analyzed by flow cytometry. (A) There was no significant separation between YMSM with and without asymptomatic STI as analyzed by PERMANOVA. Purple dots represent STI-positive YMSM; green triangles represent STI-negative YMSM. (B) Significant separation was seen between YMSM with and without HIV. Pink dots represent HIV-positive YMSM; teal triangles represent HIV-negative YMSM. (C) After adjusting for STI, LDM identified 17 cell subsets that were significantly (FDR<5%) different between YMSM with and without HIV. Cell subset percentages were log transformed for analyses. $HIV_{pos}$ = YMSM with HIV; $HIV_{neg}$ = YMSM without HIV.

addition, neutrophil, γδ T cell, and MAIT cell subsets were also increased in the rectal mucosa of YMSM with HIV. Among CD4+ T cell subsets, total, memory, CCR5+ memory, IFNγ+, TNFα+, and TRM cell subsets were significantly decreased in the rectal mucosa of YMSM with HIV. In addition, the percentage of CD56+CD16- NK cells was also decreased in the rectal mucosa of YMSM with HIV.

## Compared to YMSM without STI, the microbiome composition of YMSM with asymptomatic bacterial rectal STI demonstrated a higher probability of presence of taxa with pathogenic potential and decrease in commensal gut flora

Rectal mucosal swabs were collected during rigid sigmoidoscopy for 16S rRNA sequencing of the gut microbiome (n = 102). Similar to cellular analyses, there was no significant interaction identified between HIV and STI for microbiome outcomes, so stratified analyses were not conducted. After adjusting for sequencing batch effect and STI, there were no significant differences between the microbiome compositions of YMSM with and without HIV in alpha diversity (Chao1 p = 0.3, Shannon p = 0.09), beta diversity (Bray-Curtis p = 0.3, Jaccard p = 0.05), global relative abundance (LDM $p_{adj}$ = 0.2), or presence-absence ($p_{adj}$ = 0.05). After adjusting for sequencing batch and HIV, there were also no significant differences between the microbiome compositions of YMSM with and without STI in alpha (Chao1 p = 0.8, Shannon p = 0.8) or beta diversity (Bray-Curtis p = 0.08, Jaccard p = 0.1). When analyzing taxa-level

outcomes by LDM, relative abundance and presence-absence results differed between YMSM with and without STI in expected ways; the genus *Neisseria* was significantly more abundant ($p_{adj}$ = 0.002), and both *Neisseria* ($p_{adj}$ = 0.001) and *Chlamydia* ($p_{adj}$ = 0.003) genera had higher probabilities of presence in YMSM with STI.

We noted that a small number of participants demonstrated discordant *Chlamydia* or *Neisseria* genera detected on the 16S rRNA sequencing results as compared to the clinical diagnostic assay (n = 16). We thus reexamined the data by limiting our analyses to the specimens with *Chlamydia* or *Neisseria* genera detected on both the clinical assay and 16S rRNA sequencing (n = 86) in order to focus on patients that were most likely to represent true, clinically relevant infection. Once again, there was no significant interaction between HIV and STI, and we did not see significant effects of HIV on the microbiome after controlling for sequencing batch and STI (S2 Fig). However, we did find additional significant effects of STI on the microbiome while adjusting for batch and HIV in this restricted sample set (Fig 3). Alpha diversity metrics (Chao1 p = 0.6, Shannon p = 0.7) were not significantly different between YMSM with and without STI, but beta diversity by both Bray-Curtis (p = 0.03) and Jaccard (p = 0.02) similarity measures were significantly different between YMSM with and without STI (Fig 3C & 3D). The LDM based on global relative abundance data was also significantly different ($p_{adj}$ = 0.03) between the two groups, and *Neisseria* remained the only genus that had a significantly higher relative abundance ($p_{adj}$ = 0.003) in YMSM with STI. The global LDM statistic based on

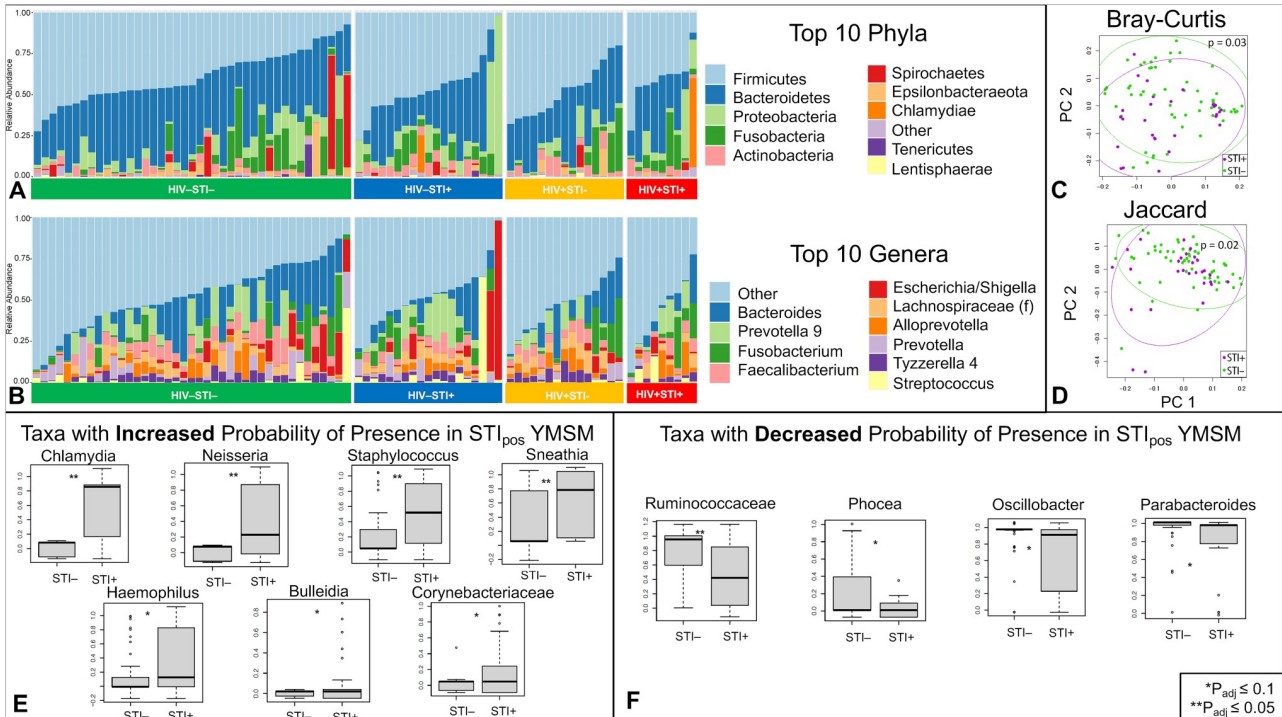

**Fig 3. Rectal mucosal microbiome composition in YMSM with and without STI.** (A) Representative bar plot of the top 10 phyla identified by 16S rRNA sequencing from rectal mucosal specimens of the 86 participants with concordant clinical diagnostic assay and 16S microbiome sequencing STI results. (B) Representative bar plot of the top 10 genera identified by 16S rRNA sequencing from rectal mucosal specimens of the 86 participants with concordant clinical diagnostic assay and 16S microbiome sequencing STI results. (C) Bray-Curtis and (D) Jaccard measures of beta diversity, both measured by PERMANOVA, both demonstrated significant differences between the microbiome composition of YMSM with and without STI. Purple dots represent STI-positive YMSM; green dots represent STI-negative YMSM. (E & F) Presence-absence analyses identified 11 genera with a significantly different (FDR<10%) probability of presence between the two groups. STIpos = YMSM with asymptomatic rectal bacterial STI; STIneg = YMSM without asymptomatic rectal bacterial STI.

presence-absence analyses was also significant ($p_{adj}$ = 0.04). In contrast to the initial evaluation of the full dataset, this re-evaluation of the limited concordant dataset identified 11 genera with a significantly different ($p_{adj}$<0.1) probability of presence between YMSM with and without STI. As expected, the genera *Neisseria* and *Chlamydia* again had a higher probability of presence among YMSM with STI controlling for sequencing batch and HIV. Five other potentially pathogenic bacteria—*Sneathia*, *Staphylococcus*, *Bulleidia*, *Haemophilus*, and unspecified genera from the *Corynebacteriaceae* family—also had a higher probability of presence in YMSM with STI. Unspecified genera from the *Ruminococcaceae* family as well as the genera *Parabacteroides*, *Oscillibacter*, and *Phocea*, all of which are commensal gut bacteria, had a lower probability of presence in YMSM with STI (Fig 3E & 3F). Of note, while there were no differences between the four study groups detected, two of the microbiome specimens from YMSM with STI and without HIV (one with GC and one with CT) appeared to be dominated by the *Shigella* genus. Examination of cellular subsets did not show these two specimens to be clear outliers, and additional analyses are ongoing given the relatively high prevalence of this taxa in our cohort overall.

## Asymptomatic bacterial STI was associated with differential expression of inflammatory genes and pathways only in the rectal mucosa of YMSM with HIV

Total RNA was extracted from rectal pinch biopsies and utilized for bulk transcriptome analyses from 43 participants in the cohort. In contrast to our cellular subsets and microbiome analyses, in this case, the LDM identified evidence of significant interaction between HIV and STI in gene expression profiles of the rectal mucosa (p = 0.008), revealing a differential effect of bacterial STI on the rectal mucosal transcriptome between YMSM with and without HIV. This interaction was attributed to 47 gene transcripts. Due to the significant interaction identified, we next conducted stratified analyses of the four study groups with DESeq2 and visualized the differences in study groups by volcano plot (Fig 4).

Among YMSM without STI, 23 genes were significantly ($p_{adj}$<0.1) differentially expressed between YMSM with and without HIV (Fig 4A), five of which (HSPB7, B3GNT3, AHNAK2, TMEM47, and COL28A1) were also among the 47 significant interaction genes identified. Among YMSM with STI, 23 genes were significantly ($p_{adj}$<0.1) differentially expressed between YMSM with and without HIV (Fig 4B), six of which (DMBT1, C4BPA, C4BPB, ZNF132, CASP5, and TNIP3) were also among the 47 significant interaction genes. The genes that were significant in both the global interaction and stratified analyses all contribute to immune function and regulation [14–18] or tissue homeostasis [19–22].

We then stratified by HIV status and examined the effect of STI on the rectal mucosal transcriptome. Among YMSM without HIV, no genes were significantly differentially expressed in the rectal mucosa between YMSM with and without STI (Fig 4C). Among YMSM with HIV, however, asymptomatic bacterial STI was associated with significant upregulation of 323 and downregulation of 109 genes (Fig 4D), which is nearly 19 times more differential expression than was seen when evaluating the effect of HIV on YMSM with and without STI. Forty of the 47 significant interaction genes from the global interaction analysis were also significant in this stratified analysis (Fig 4E). Nearly all of these genes are involved in inflammation (e.g., CASP5, MAPK6, TRIM40, TRIM25, TNIP3, PLAUR, C4BPA, C4BPB, TNFSF10, B3GNT3, DMBT1, XDH, IRS2, UPP1, NT5C3A, CFLAR, MXD1, SEMA4B, ST3GAL4, PLAC8, CEACAM5), cell migration (e.g., RAPH1, ARL14, BAIAP2L1), or mucosal integrity (e.g. MUC13, CLDN23, ANGPTL4, FHL2, KIAA1211, ABHD11-AS1, LMO7, HK2).

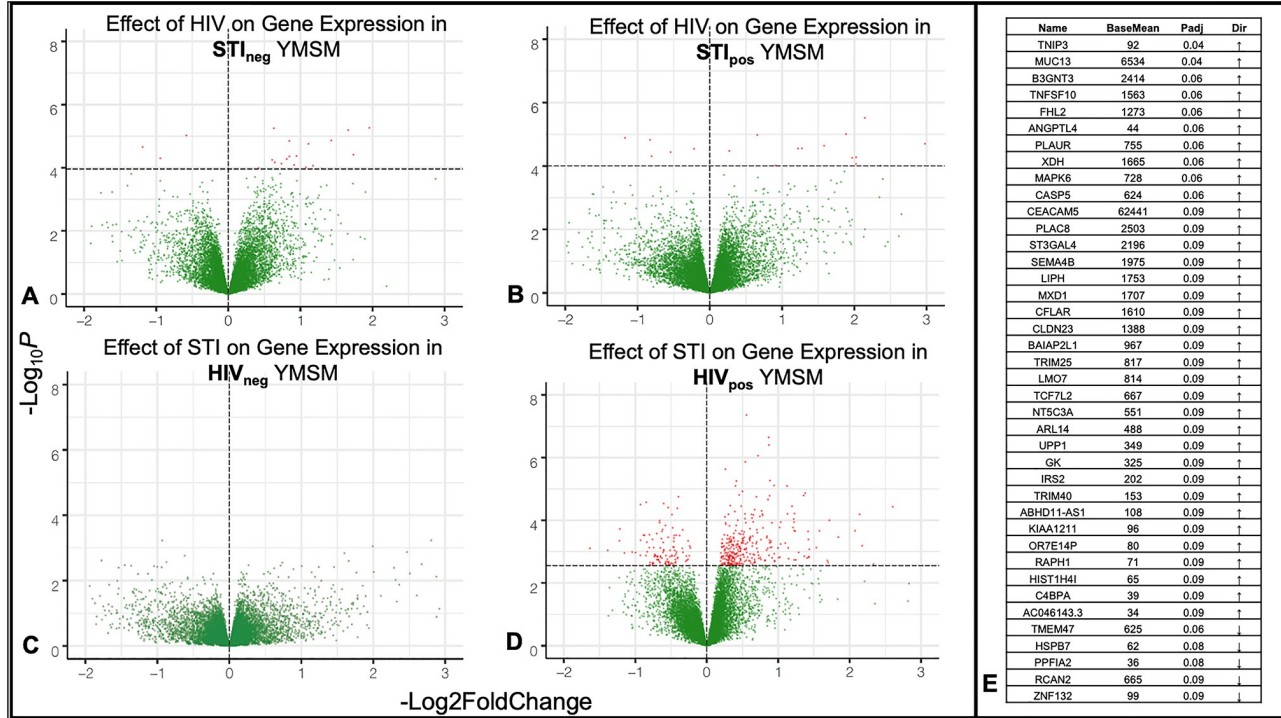

**Fig 4. Asymptomatic bacterial STI is associated with differential gene expression among YMSM with HIV only.** (A-D) Volcano plots of differentially expressed genes in the four study groups. Red dots indicate genes with significant ($p_{adj} < 0.1$) differential expression. (E) List of 40 genes that were identified both as interaction genes and as differentially expressed among YMSM with HIV and rectal asymptomatic bacterial STI. STIpos = YMSM with asymptomatic bacterial STI; STIneg = YMSM without asymptomatic bacterial STI. $HIV_{pos}$ = YMSM with HIV; $HIV_{neg}$ = YMSM without HIV.

To further characterize the differential effect of asymptomatic STI on the rectal mucosal transcriptome between YMSM with and without HIV, Gene Set Enrichment Analyses (GSEA) were then undertaken with the Molecular Signatures Hallmark and Reactome Pathway Databases [23]. These analyses demonstrated significant differential upregulation of many important inflammatory pathways within the rectal mucosa of YMSM with HIV and STI (Fig 5) that was not apparent in the rectal mucosa of YMSM without HIV and with STI, further supporting a distinctive effect of asymptomatic bacterial STI based on HIV status. Because only a few genes were differentially expressed in our stratified analyses evaluating the effect of HIV in YMSM with and without STI, we did not pursue further pathway analyses.

### There was no significant effect of asymptomatic bacterial STI on *in vivo* or *ex vivo* HIV replication in rectal tissues

For YMSM with HIV, HIV-1 viral RNA was quantified in both rectal biopsy tissue and serum specimens. We compared rectal and serum viral loads between YMSM with HIV and with and without STI using a one-sided t-test and found no significant difference between these groups. For YMSM without HIV who were not taking PrEP, we performed rectal mucosal *ex vivo* explant challenge experiments and compared p24 production in the supernatant over 18 days between YMSM with and without STI. The median log area under the curve ($_{log}$AUC) was calculated for each YMSM without HIV, and no significant difference was found in the $_{log}$AUCs between those with and without STI (Fig 6).

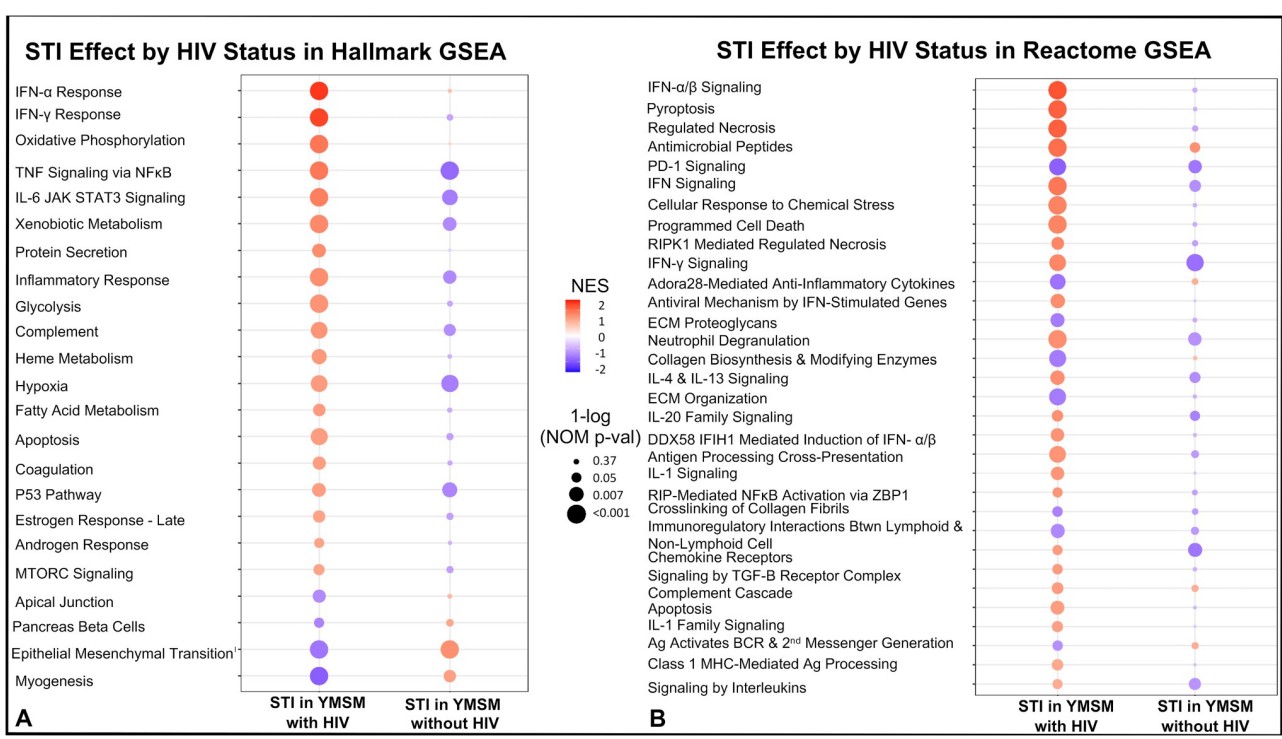

**Fig 5. Asymptomatic bacterial STI is associated with upregulation of transcriptomic pathways involved in inflammation and mucosal injury and repair among YMSM with HIV.** Gene set enrichment analyses were conducted using the (A) Hallmark gene set and (B) Reactome gene set. Red dots represent pathways that are upregulated and purple dots represent pathways that are downregulated. The size of the dot corresponds to the nominal p-value.

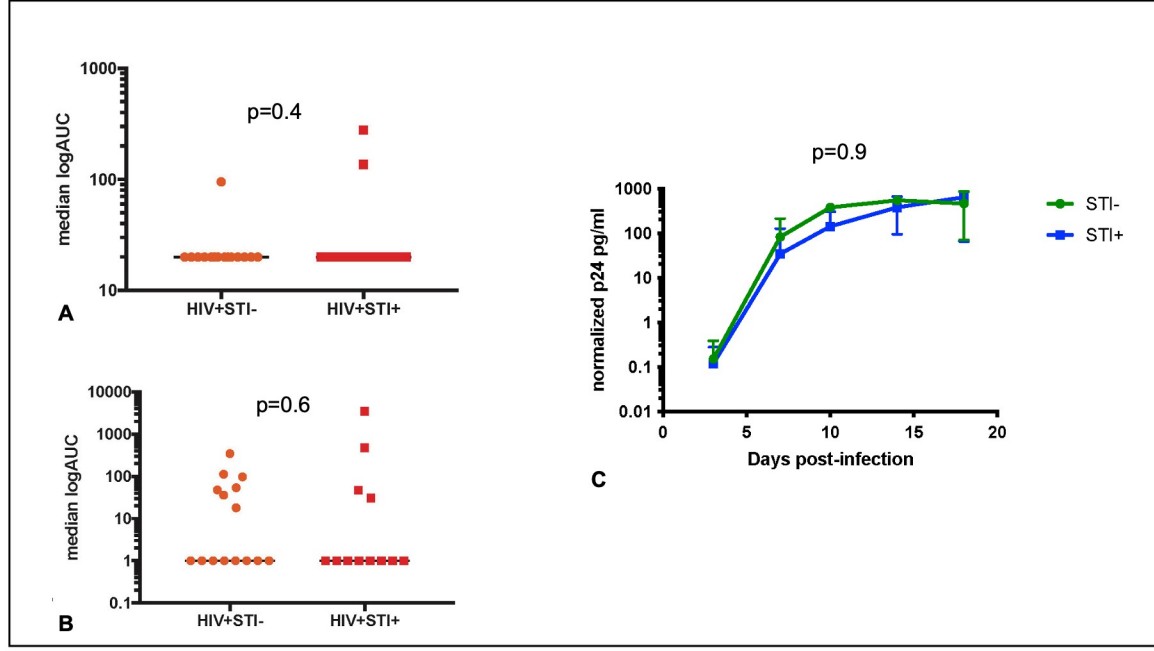

**Fig 6. There is no significant effect of asymptomatic rectal STI on HIV replication in blood or rectal tissues.** HIV viral loads in (A) plasma and (B) rectal tissues among YMSM with HIV. Orange circles represent YMSM with HIV and without STI. Red squares represent YMSM with HIV and with STI. (C) Plot of p24 concentrations from rectal explant challenge experiments among YMSM without HIV and not on oral PrEP. Green circles represent YMSM without HIV and without STI. Blue squares represent YMSM without HIV and with STI.

## Discussion

Here, we have comprehensively characterized the rectal mucosal immune effects of the syndemic infections of HIV and asymptomatic bacterial STI that disproportionately affect YMSM. We found profound alterations in the rectal mucosal cellular composition among YMSM with HIV despite suppressive ART and preserved systemic CD4+ T cell counts. While we hypothesized that asymptomatic bacterial STI would be associated with rectal mucosal inflammation in both YMSM with and without HIV, our data showed differential effects based on HIV status in the rectal mucosal transcriptome. We also showed that the presence of asymptomatic rectal GC and/or CT was associated with increased probability of presence of other pathogenic taxa in the microbiome. The additive effects of the HIV-related rectal mucosal cellular composition changes plus the STI-related microbiome shift could have contributed to our finding of rectal mucosal inflammation in the transcriptome among YMSM with both HIV and asymptomatic bacterial STI.

In the rectal mucosa of YMSM with HIV, we saw rectal mucosal enrichment of CD8+ T cells, Ki67+ CD4+ and CD8+ T cells, γδ T cells, MAIT cells, and neutrophils and depletion of CD4+ and CD4+ CCR5+ T cells, which is consistent with prior research [24–29]. We also found that CD69-CD103- (non-tissue-resident) CD8+ T cells were enriched and CD56 +CD16- NK cells and CD69+ CD4+ T cells were depleted in YMSM with HIV. These findings highlight that HIV perturbs components of both the innate and adaptive immune systems in the gut cellular compartment even in the setting of effective ART [24–27,29,30]. It is also notable that all participants in this study were healthy, relatively young, and on suppressive ART with preserved peripheral CD4 T cell counts. While others have also observed, for example, NK depletion within the gut in an older cohort of people with HIV (median age of 47 years), this was only in the absence of antiviral therapy [30]. Significant depletion of gut NK was evident in our cohort of younger individuals on ART. We did not identify effects on the cellular composition of the rectal mucosa associated with asymptomatic bacterial STI regardless of HIV status, which was a somewhat unexpected result. Different results might be seen in the setting of symptomatic proctitis associated with STI, but the immunologic effects of asymptomatic bacterial STI may be too subtle to detect by flow cytometry and may require more sensitive methods.

Evidence of STI-associated inflammation in the rectal mucosal transcriptome of YMSM was only apparent among YMSM with HIV. Our findings could have implications for the increased risk of chronic inflammation-related disease associated with HIV and highlight the essential contributions of the syndemic research approach [31]. This may be particularly relevant for YMSM with HIV and recurrent rectal STI, which is a frequent clinical experience [32–34]. Our study showed upregulation of the gene PLAUR in YMSM with HIV and asymptomatic bacterial STI. The soluble receptor that this gene codes for, suPAR, is a biomarker of inflammation seen in numerous disease states. High suPAR levels have been associated with non-AIDS events in people with HIV [35]. C4BP, another gene of interest shown to be upregulated in the rectal mucosa of YMSM with HIV and asymptomatic bacterial STI, has previously been shown to be elevated in people with HIV and represents a proposed mechanism for the increased thrombosis risk in HIV [36]. CASP5 was also enriched among YMSM with HIV and asymptomatic bacterial STI; caspase cascades are an important mediator of inflammatory responses and have been implicated in HIV-associated neurodegeneration [37]. Our pathway analyses further supported an enhanced inflammatory response in the setting of asymptomatic bacterial STI among YMSM with HIV compared to YMSM without HIV and suggest this should be explored as a contributor to chronic inflammation in people with HIV.

Our findings of increased inflammation associated with asymptomatic STI among YMSM with HIV may also have important implications for HIV cure efforts as inflammation is thought to entrench the HIV reservoir [11], and gut reservoir entrenchment in particular is a major barrier to cure efforts [38]. In YMSM with HIV and asymptomatic bacterial STI, we found upregulation of several genes with potential roles in HIV reservoir maintenance including those associated with antiviral immunity, HIV replication, and T cell homeostasis such as TRIM25, TRIM40, MAPK6, and TNFSF10 [39–44]. While we did not detect a difference in tissue HIV viral loads in our study, future research will be necessary to fully understand the effects of asymptomatic rectal STI on the gut HIV reservoir in people with HIV.

We showed that asymptomatic bacterial STI in YMSM without HIV is not associated with transcriptional or immunologic markers of rectal mucosal inflammation, which represents a significant contribution to the current knowledge of rectal STI pathogenesis. Prior studies have assessed *in vitro*, cervical, and urethral inflammation in the setting of bacterial STI [45–47], but rectal mucosal inflammation attributed to bacterial STI has been understudied to date despite the tremendous burden of rectal STI. Broadly, mucosal GC and/or CT infection have been associated with enhanced cervical Th2 response [46] and higher urethral IL-8 [46] as well as IL-6 and TNFα [47] in the male and female genital tract. One prior study of asymptomatic rectal CT infection among MSM with and without HIV also showed no evidence of enhanced inflammation with a limited number of biomarkers assessed, which is overall consistent with our data [48]. Given our limited sample size and the presence of co-infections in our cohort, we were unable to examine the individual effects of asymptomatic GC, CT, or syphilis, and it is possible that inflammatory effects differ between these asymptomatic infections.

Asymptomatic bacterial STI was associated with a microbiome shift away from commensal gut flora and towards detection of potential pathogens regardless of HIV status in our study. *Sneathia*, which was enriched among YMSM with bacterial STI in our study, presence in the vaginal microbiome has been associated with adverse reproductive and perinatal outcomes, and prior studies have demonstrated an association between *Sneathia* colonization and both vaginal STIs and HIV [49]. The *Staphylococcus* and *Haemophilus* genera as well as unspecified genera from the *Corynebacteriaceae* family were also enriched among YMSM with bacterial STI and are capable of causing clinical infection if they gain access to sterile sites via compromised membranes or structures. Gastrointestinal *Staphylococcus aureus* carriage, in particular, has been associated with subsequent systemic infection [50].

Generally, pathogenic colonization of the gastrointestinal tract and displacement of commensal gut bacteria can instigate chronic inflammation by increasing the pro-inflammatory potential of the intestinal microbiota [51], and the gut microbiome has been shown to modify the bioactivity of rectal toll-like receptor ligands such as LPS [52,53]. The only other known study of the effect of STI on the rectal microbiome found a shift towards anaerobic taxa in the presence of STI [54], but this study differed from ours in that it included individuals with symptomatic STI (18–65% depending on the STI) who were older than our study population (mean age in the 30s).

We did not detect an effect of HIV on the microbiome in this relatively homogenous population of YMSM. In prior studies, the effects of HIV on the microbiome have been inconsistent due to various confounding factors including sexual behaviors [55]. Other clinical factors absent in our study participants, such as having acute HIV infection [56], advanced HIV disease [57], low CD4 nadir [58], suboptimal immune recovery on ART [59], older age [60], and/ or men who have sex with women (MSW) [61] have been associated with microbiome

differences among people with HIV. We hypothesize that the young age, similar behaviors and demographics, preserved peripheral CD4 counts, and moderate sample size of our cohort likely contributed to the lack of differences in microbiome composition in our study.

Several studies of rhesus macaques have found that the rectal microbiome composition affects SIV acquisition [62–64]. Prior work [65,66] has also shown that asymptomatic rectal STI is associated with HIV seroconversion even in the setting of routine screening and treatment, which suggests that ongoing mucosal inflammation associated with asymptomatic STI could increase susceptibility to HIV. In our prior studies utilizing the explant challenge model to approximate HIV acquisition and early replication, we found increased p24 production was associated with differences in rectal mucosal cellular composition and the transcriptome [67]. Given the lack of immunologic effects of asymptomatic bacterial STI on either the cellular composition or the transcriptome in YMSM without HIV, the lack of differences in p24 production is not surprising and suggests that prior epidemiologic associations between asymptomatic STI and HIV acquisition may be confounded by behavioral factors. We also did not observe significant differences in rectal tissue HIV viral loads in YMSM with HIV and asymptomatic bacterial STI despite evidence of inflammation in the rectal mucosal transcriptome. It is possible that participant diversity in both viral (e.g., viral diversity between participants, time of infection, peak viral load) and bacterial (e.g., bacterial diversity, time of infection, bacterial load) factors obfuscated our ability to identify a direct association between HIV replication and STI. Of note, the STI-related microbiome differences that we saw involved different taxa than those identified in the nonhuman primate studies and, thus, may not imply the same effect on SIV/HIV acquisition.

Our study sample size was modest and lacked statistical power to assess all possible immunologic associations or to stratify by STI pathogen. We also had limited clinical data on participants particularly with respect to duration of HIV, nadir CD4 count, or ART regimens. Nonetheless, we examined a large number of immunologic outcomes. We also utilized robust statistical methods to control for multiple comparisons, which could obscure some outcomes. We intentionally focused our study on asymptomatic bacterial STI as this represents the majority of infections in clinical practice. Symptomatic proctitis likely represents a different, pro-inflammatory state. Our study focused on YMSM with HIV on effective ART. Older MSM or YMSM with uncontrolled HIV likely have very different immunologic phenotypes and could have different responses to asymptomatic bacterial STI. We included diverse asymptomatic bacterial infections (GC, CT, TP, and coinfection) in this analysis due to sample size limitations. We conducted exploratory analyses to determine whether any differences were evident between the individual pathogens and saw no evidence for this in our analyses; however, statistical power to detect differences was extremely limited.

Our work contributes valuable information to the immunopathogenesis of asymptomatic bacterial STI in the rectal mucosa of YMSM and identifies an important syndemic interaction between HIV and STI in YMSM. It is essential to understand host immune responses to both HIV and asymptomatic bacterial STI, as well as the multiplicative interactions between them, to address the full spectrum of care for these conditions including interventions for prevention, treatment, and cure. Our findings provide a basis for future studies evaluating whether STI treatment alters systemic inflammation in people with HIV, whether asymptomatic bacterial STI in YMSM affects the rectal mucosal immune environment as those patients age, and whether frequent rectal STI among YMSM with HIV results in development of other co-morbidities or represents another potential barrier to HIV cure.

## Materials and methods

### Ethics statement

Ethical approval for this study has been obtained from The Institutional Review Board (IRB) at Emory University (approval numbers: IRB00092959 and IRB000094847). Written informed consent was obtained from all study participants.

### The clinical cohort

YMSM aged 18–29 years in good overall health with and without untreated asymptomatic rectal gonorrhea, chlamydia, and/or early latent syphilis were recruited from the Atlanta community regardless of HIV status. YMSM living with HIV included in this analysis were on antiretroviral therapy (ART) with plasma viral load <300 copies/ml (median<20 copies/ml) and CD4 counts >200 (median = 635). After providing written informed consent, participants underwent peripheral blood and rectal biopsy and secretion sampling via rigid sigmoidoscopy as previously described [55]. Treatment of the bacterial STI was provided after the sampling procedures. All participants were asked to abstain from receptive anal intercourse for seven days after biopsy procedures to allow the mucosa to heal and to complete the antibiotic course for YMSM with STI. All participants were educated about pre-exposure prophylaxis and HIV treatment and referred to services as needed. Please see S1 Fig for a summary of study methods.

### Blood and rectal mucosal mononuclear cell phenotyping

Peripheral blood mononuclear cells (PBMCs) were collected via CPT tubes, and six pinch biopsies from the rectal mucosa were processed by collagenase digestion to separate mucosal mononuclear cells (MMCs) as described previously [68]. Briefly, mononuclear cells isolated from the blood and rectal biopsies were stained with three staining panels to quantify immune cell subsets of interest: Panel 1 (LIVE/DEAD Aqua, CD45, CD3, CD4, CD8, CD45RA, CD69, CD103, CD25, CCR7, a4b7, CCR5, Ki67, FOXP3), Panel 2 (LIVE/DEAD Aqua, CD45, CD3, CD8, HLADR, CD20, CD16, CD56, CD66b, CD123, CD1c, CD11c, and CD14 [PBMC] or CD163 [rectal]), and Panel 3 (LIVE/DEAD Aqua, CD45, CD3, CD4, CD8, TCR g/d, TCR Va7.2, CD161, CCR5, a4b7, IL-17, TNFα, IFNγ) which was stained after stimulation in with 200 ng/ml of PMA and 1 μg/ml of Ionamycin in the presence of Brefeldin A (5 μg/ml; Sigma) and Golgi stop (0.5 μl/ml; BD Pharmingen) and 4 hour incubation at 37˚C, while γδ T cells and MAIT cells were quantified via unstimulated cells. Events were acquired on the LSR-Fortessa platform and analyzed with Flowjo software (Treestar Inc. CA). We used the gating strategies previously reported by our group for these analyses [60,67] and depicted in S3–S5 Figs.

### Microbiota sequencing

Swabs of the rectal mucosa were collected prior to biopsy through the sigmoidoscope at approximately 8–10 cm from the anal verge and stored at −80˚C. DNA was extracted, amplified, and sequenced using methods described previously [69]. Briefly, DNA is extracted using the Qiagen DNeasy PowerSoil Pro Kit. Libraries are made using a modification of the Illumina 16S Metagenomic Sequencing Library Preparation workflow. 12.5 ng of DNA is amplified using 16S Amplicon PCR Forward and Reverse Primers. Libraries are purified with Ampure XP beads. Purified amplicons are indexed with Nextera XT Index primers, and indexed amplicons are purified with Ampure XP beads. Final 16S libraries are approximately 630 bp and are pooled in equal amounts based on fluorescence quantification. Final library pools are quantitated via qPCR. The pooled library is sequenced on an Ilumina MiSeq using MiSeq v3 600

cycle chemistry at a loading density of 6–8 pM with 20% PhiX, generating roughly 20 million, 300 bp paired-end reads. Recommended sequencing depth is >100,000 reads per sample. Raw amplicon sequence reads were evaluated for quality control (QC) using the FastQC suite with MultiQC [70,71] and were then processed using Quantitative Insights into Microbial Ecology (QIIME2 v2021.2) [72]. The Divisive Amplicon Denoising Algorithm 2 (DADA2) package [73] was used within QIIME2 to denoise and dereplicate all paired-end sequences and to create the feature table of amplicon sequence variants used within QIIME2. DADA2 parameters were chosen to trim the first 30 bp and to truncate both paired-end reads at position 240. Taxonomic assignment was performed via QIIME2 and the data were aligned to Silva (v132) [74] using the QIIME taxonomy modules.

## Transcriptome sequencing

RNAseq analyses were conducted at the Emory Primate Research Center NHP Genomics Core Laboratory (http://www.yerkes.emory.edu/nhp_genomics_core/). Two rectal biopsies from each participant were placed in RNALater (Invitrogen, #AM7021) and stored at -80˚C until processing. Both biopsies were homogenized in 350 mL of Buffer RLT and then extracted using the RNeasy Micro kit (Qiagen) with on-column DNase digestion. RNA quality was assessed using a Bioanalyzer (Agilent), and then ten nanograms of total RNA was used as input for cDNA synthesis using the Clontech SMART-Seq v4 Ultra Low Input RNA kit (Takara Bio) according to the manufacturer's instructions. Amplified cDNA was fragmented and appended with dual-indexed bar codes using the NexteraXT DNA Library Preparation kit (Illumina). Libraries were validated by capillary electrophoresis on a TapeStation 4200 (Agilent), pooled at equimolar concentrations, and sequenced on an Illumina HiSeq3000 at 100SR, yielding ~18 million reads per sample. Alignment was performed using STAR version 2.5.2b [75] using default parameters and transcripts were annotated using GRCh38. Transcript abundance estimates were calculated internal to the STAR aligner using the algorithm of htseq-count [76] in union mode.

## *Ex vivo* explant challenge experiments

Three biopsies from each participant without HIV were individually weighed and exposed to HIV-1 BaL ($10^{2.8}$ $TCID_{50}$ in a volume of 250 µl media) for 2 h (37˚C, 5% $CO_2$). Each biopsy was then extensively serial-washed in sterile PBS (5 x 500 µl) and placed on a pre-soaked, pre-warmed collagen raft (Ethicon Surgifoam, #1972) in 1 ml complete media (RPMI 1640 with 10% FBS, Gentamicin Sulfate, and Zosyn). At set post-infection timepoints (Day 3, 7, 10, 14, 18), 700 µl media was removed and replaced with 700 µl complete media. Collected supernatants were stored at -30˚C until p24 analysis. P24 was quantified via ELISA (ABL, Inc. #5447) according to the manufacturer's instructions. P24 values were normalized to biopsy weight.

## HIV viral load on rectal biopsy specimens

Two rectal biopsies from each participant with HIV were weighed prior to being placed in Promega lysis buffer for overnight tissue digestion. HIV-1 RNA was quantitated using the Abbott Realtime HIV-1 Assay procedure. Results are reported as copies per milligram of tissue. Biopsies with results below the assay limit of detection were assigned values of 1.

## GC/CT and syphilis detection

The Abbott Real Time CT/NG PCR was utilized to detect *Neisseria gonorrhea* and/or *Chlamydia trachomatis* from rectal swabs collected during rigid sigmoidoscopy. RPR with Reflex

Titer and Fluorescent Treponemal Antibody Confirmatory Testing was utilized to detect Treponema pallidum from blood samples at a commercial laboratory (Quest Diagnostics).

## Statistical analyses

Demographic and sexual behavior characteristics were compared between study groups using the Wilcoxon rank-sum test for continuous variables and the chi-squared or Fisher's exact test for categorical variables, as appropriate.

## Immune cell subset statistical analyses

Given our modest sample size and multiple parameter outcomes of interest, we first used discovery-based linear decomposition modeling (LDM) that allowed for inclusion of all four study groups into a single model, controlled for multiple comparisons, and is a single analysis pathway that combines both global tests of effect as well as tests of individual outcomes [13,77]. LDM also enables the inclusion of interaction terms in the model to assess for effect modification. In this case, we were interested in interaction between HIV and STI, meaning that the effect of STI on immune cell subsets could differ based on the participant's HIV status. To visualize global differences, two-dimensional principal coordinates analyses (PCoAs) were constructed. We then used LDM to produce individual p values for assessing differences of individual cell subsets between groups, and adjusted p values that account for multiple comparisons while controlling for FDR at nominal level 10%. The LDM essentially fits a linear model for the cellular subset data as the outcome and regresses it on continuous traits or categorical group variables while adjusting for potential confounders; however, it differs from standard linear regression in that it uses permutation-based p values to account for non-normally distributed cellular subset data. For these analyses, data for each cell subset were normalized to have mean zero and standard deviance of one; missing data were imputed by the mean of the observed data in the same cellular subset.

## Microbiome statistical analyses

We began our analyses of microbiome composition by using the above-described discovery-based linear decomposition modeling (LDM) to look for interaction between HIV and STI for the assessed outcomes of alpha and beta diversity, relative abundance, and presence-absence data. LDM is not compositionally aware by design, but we elected to use it in order to avoid some inherent biases in compositionally aware methods, such as the biases associated with the introduction of pseudo counts and the incapability of handling rare taxa. Taxonomic alpha diversity, or the distribution of species abundances in a given sample, was estimated using the Chao1 and Shannon indices and measured with a Wilcoxon test. Beta diversity, or dissimilarity in community composition, was estimated using the Bray-Curtis and Jaccard indices and measured using PERMANOVA. This was calculated using an ASV table and with vegan software package. Relative abundance provides the percentage of the microbiome that is made up of a specific organism and was analyzed by LDM. Presence-absence analysis assesses the probability that a taxon is present within a given sample and is the preferred analysis for identifying rare taxa. Many associations are driven by changes in which taxa are present and which are absent, however confounding by read depth is a limitation to this analysis. Our method of evaluating presence-absence involves first rarifying the ASV table such that all samples have the same library size, which eliminates confounding by read depth, and then repeatedly applying the LDM to all rarified taxa count tables [77]. Beta diversity was visualized with Principal Coordinates Analysis (PCoA). We first generated a global p value with the LDM and then

assessed associations with individual taxa. No interaction was found via any of these measures, and so stratified analyses were not indicated. All analyses were conducted at the genus level.

## RNA-seq statistical analyses

We restricted our analyses to those samples run in a single sequencing plate in order to avoid the risk of sequencing batch effect. The single batch we selected included the majority of participants in our four study groups with representation from all groups included on the same plate. After RNA sequencing read alignment, gene-wise and isoform-wise expression levels for individual genes were estimated using the program DESeq2, which normalizes gene expression level estimates across samples and also corrects for non-uniformity in read distributions across each gene [78]. We again used the LDM to test for interaction and examine global as well as individual gene level differences between the groups. Differentially expressed genes were tested for enrichment in gene families/pathways/protein interactions using Gene Set Enrichment Analysis with the Immunological Signatures Databases hosted at the MSigDB database (www.broadinstitute.org/gsea/msigdb/collection.jsp). GSEA was performed on the regularized log (rlog) expression table produced by DESeq2 employing a weighted enrichment statistic and Signal2Noise as the ranking metric and using 1000 phenotype permutations. Because there is no consensus on gene normalization or interaction correction for PCoA analysis, we did not construct PCoA plots for our transcriptome data.

## Explant challenge and rectal mucosal HIV viral load statistical analyses

For participants without HIV, the median logAUC p24 values and the p24 concentrations, normalized by biopsy weight, from each day of assessment were compared between study groups by the Kruskal-Wallis test with Dunn's correction. For participants with HIV, the median rectal viral loads were compared between study groups by the Mann Whitney t test.

## Supporting information

**S1 Table. Immune cell subsets characterized by flow cytometry on blood and rectal mucosal (RM) samples.**
(DOCX)

**S1 Fig. Summary of study methods.** Each study participant underwent rigid sigmoidoscopy for the collection of study specimens. Rectal mucosal swabs were collected for 16S rRNA microbiome sequencing. Six rectal mucosal pinch biopsies were collected for immune cell quantification via flow cytometry. Two rectal mucosal pinch biopsies were collected for RNA-seq transcriptomic analysis. For YMSM with HIV, two rectal mucosal pinch biopsies were collected for HIV viral load detection. For YMSM without HIV, three rectal mucosal pinch biopsies were collected for our *ex vivo* HIV challenge. Figure created with BioRender.com.
(TIFF)

**S2 Fig. Rectal mucosal microbiome composition in YMSM with and without HIV.** (A) Bray-Curtis and (B) Jaccard measures of beta diversity both demonstrated no significant differences between the microbiome composition of YMSM with and without HIV. Pink dots represent HIV-positive YMSM; teal dots represent HIV-negative YMSM.
(TIFF)

**S3 Fig. Representative gating strategy for rectal mucosal mononuclear cells.** Lymphocytes were identified by forward and side scatter. CD45+ cells were then isolated, followed by CD3+ cells which were separated into CD4+ and CD8+ subsets. Memory CD4 cells were identified

by excluding CD45RA+ cells. CD69 marker was then used to divide CD4+ cells into tissue resident and non-tissue resident populations. Memory CD8 + cells were designated as being CCR7⁻ and CD45RA$^{+/-}$. Among memory CD8+ T cells, both CD69 and CD103 markers were used to designate tissue resident populations. Memory CD4+ populations, including tissue resident and non-tissue resident subsets, were then assessed for expression of CCR5, α4β7, and Ki67.
(TIFF)

**S4 Fig. Representative gating strategy to detect cytokine-positive rectal mucosal CD4+ and CD8+ T cells.** Rectal CD4+ and CD8+ MMCs were stimulated for 4 hours with PMA/Ionomycin and stained for indicated cytokines. Live cells were identified by live/dead staining and lymphocytes were identified by forward and side scatter. CD45+ cells were then isolated, followed by CD3+ cells which were separated into CD4+ and CD8+ subsets. Stimulated CD4+ T cells were assessed for IL-17A, IFNγ, and TNFα cytokine production and stimulated CD8+ T cells for IFNγ and TNFα. Abbreviations: MMC, mucosal mononuclear cells, PMA, phorbol myristate acetate, IFNγ, interferon gamma, TNFα, tumor necrosis factor alpha.
(TIFF)

**S5 Fig. Representative gating strategy to detect neutrophils, NK, macrophage, and B cells.** Representative flow panel to identify non-T-cell immune cell subsets: Lymphocytes were identified via forward and side scatter, singlets, and live cells. Neutrophils were CD66+ CD45 + cells. CD45+CD3-CD20- cells were further divided by negative gating to CD56+ and CD56 +CD16+ NK, macrophages by CD16, CD163, and HLA-DR expression, while CD1c+ and pDC were identified with CD11c, CD1c, and CD123. B-cells were defined by HLA-DR and CD20 expression.
(TIFF)

## Acknowledgments

We express our sincerest gratitude to the study participants. We also gratefully acknowledge the Hope Clinic clinical research and community engagement team.

## Author Contributions

**Conceptualization:** Colleen F. Kelley.

**Data curation:** Vanessa E. Van Doren, S. Abigail Smith, Yi-Juan Hu, Gregory Tharp, Steven Bosinger, Cassie G. Ackerley, Phillip M. Murray, Rama R. Amara, Praveen K. Amancha, Robert A. Arthur, H. Richard Johnston.

**Formal analysis:** S. Abigail Smith, Yi-Juan Hu, Gregory Tharp, Steven Bosinger, Robert A. Arthur, H. Richard Johnston.

**Funding acquisition:** Colleen F. Kelley.

**Investigation:** S. Abigail Smith, Phillip M. Murray, Rama R. Amara, Praveen K. Amancha.

**Methodology:** S. Abigail Smith, Yi-Juan Hu, Gregory Tharp.

**Project administration:** S. Abigail Smith, Cassie G. Ackerley, Phillip M. Murray, Colleen F. Kelley.

**Resources:** Colleen F. Kelley.

**Supervision:** S. Abigail Smith, Cassie G. Ackerley, Colleen F. Kelley.

**Writing – original draft:** Vanessa E. Van Doren.

**Writing – review & editing:** Vanessa E. Van Doren, S. Abigail Smith, Gregory Tharp, Cassie G. Ackerley, Robert A. Arthur, Colleen F. Kelley.

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
