## [Decision Letter · Decision Letter 0]

20 Mar 2023

Dear Dr Van Doren,

Thank you very much for submitting your manuscript "HIV, asymptomatic STI, and the rectal mucosal immune environment among young men who have sex with men" for consideration at PLOS Pathogens. As with all papers reviewed by the journal, your manuscript was reviewed by members of the editorial board and by several independent reviewers. The reviewers appreciated the attention to an important topic. Based on the reviews, we are likely to accept this manuscript for publication, providing that you modify the manuscript according to the review recommendations.

Sincerely,

Daniel C. Douek

Academic Editor

PLOS Pathogens

Susan Ross

Section Editor

PLOS Pathogens

Kasturi Haldar

Editor-in-Chief

PLOS Pathogens

orcid.org/0000-0001-5065-158X

Michael Malim

Editor-in-Chief

PLOS Pathogens

orcid.org/0000-0002-7699-2064

Reviewer Comments (if any, and for reference):

Reviewer's Responses to Questions

**Part I - Summary**

Reviewer #1: Comments to authors:

Here the authors have explored how HIV, other sexually transmitted diseases, and the composition of the GI tract bacterial microbiome contribute to inflammation in young men who have sex with men. The work is extremely important, considering that risk factors for HIV acquisition are known to influence the composition of the microbiome, and accumulating data suggest that these can increase susceptibility to acquisition. The authors use next generation sequencing and flow cytometry to assess the transcriptomics and immune cell phenotypes of the host and the composition of the bacterial microbiome. The authors find that asymptomatic HIV does not influence the composition of the microbiome, but other STIs are associated with dramatic bacterial dysbiosis. Consistent with many previous reports, the authors find that asymptomatic HIV infection is associated with dramatic immunological perturbations, but the authors also find that other asymptomatic STIs are also associated with inflammation. The authors argue their findings have important implications for how to clinically manage individuals with risk factors for HIV acquisition. Overall, the experiments are well performed, appropriate controls are included, and the conclusions are supported by the data presented.

Reviewer #2: In this study, researchers studied YMSM with and without HIV and asymptomatic bacteria STI (gonnorrhea, chlamydia, and syphillus, etc) to understand potential interactions of these infections on the rectal mucosal immune environment. They assayed 7 innate and 19 adaptive immune populations in the rectum with flow cytometry, RNASeq and also evaluated the mucosal microbiome with 16S rRNA sequencing. They describe strong alterations in immune cell populations in rectum of YMSM with HIV, which is a strong contribution to the field since they had pretty in depth immune data. They do not detect microbiome differences with HIV, which is not entirely surprising since HIV-associated microbiome differences have been detected but subtle in prior studies that controlled for MSM. Perhaps the most interesting finding is in their RNASeq data, where they describe an interaction in which bacterial STI was associated with upregulation of inflammatory genes and enrichment of immune response pathways in YMSM with HIV but not without HIV. This result was quite robust, and this has implications for treating concurrent STIs in YMSM, and understanding potential drivers of chronic inflammation observed in PLWH. Overall the paper is well written. With 105 study participants, the cohort size is decent but a little unbalanced so some cohorts are still quite small(e.g. only 14 with HIV and STI and 48 without either). It also has the weakness that they did not have the power to stratify by STI, and instead binned people with Gonorrhea, Chlamydia, and Syphilus into a single “with bacterial STI” group but these are very different pathogens that might influence very different immune subsets/ levels of inflammation. These caveats are appropriately discussed in the Discussion. The paper in places could use better description of the methods employed and other minor revisions as detailed below, but overall it is a strong paper.

Reviewer #3: The authors present a study examining relationships between HIV and STI’s with respect to mucosal microbiomes, immune cell proportions, gene expression, and ex vivo HIV infection susceptibility. The study is well-designed and provides unique insights into the interactions between HIV and STIs in several highly relevant biological systems. Inflammation in HIV is associated with all-cause mortality and numerous comorbidities, though the etiology of this ongoing inflammation remains poorly understood. The authors present data suggesting the combination of STIs and HIV may uniquely contribute to this ongoing inflammation in a way that does not occur in people without HIV.

**Part II – Major Issues: Key Experiments Required for Acceptance**

Reviewer #1: (No Response)

Reviewer #2: none

Reviewer #3: 1.What was the median and range of time since HIV diagnosis for these individuals? And time on ART? While confounding effects from MSM status have caused inconsistent results across studies, several studies have been performed that matched HIVpos and HIVneg for MSM status. Some of these studies show no significant differences in the gut microbiome of HIVpos individuals when looking early in infection (10.1016/j.ebiom.2022.104286), but studies of MSM-matched chronically-infected adults show significant differences compared to HIVneg (DOI:10.1038/s41598-018-32585-x, 10.1038/s41467-020-16222-8, DOI:10.1038/s41598-017-06675-1). It will be helpful for readers to know if these HIVpos are early in their infection and whether that may be why no differences in the microbiome are seen.

2.An aspect of the analytical design makes it difficult to confirm the authors’ conclusion that HIV has a greater impact on immune cell subsets than STIs. When HIVpos are compared to HIVneg, the difference between them is one infectious agent that causes its own unique skewing of immune subset proportions. When comparing STIpos to STIneg, several infections are grouped which may have different effects on immune cell subsets, making groupwise comparisons problematic. Groupwise comparisons test whether a group as a whole differs in a common, concerted way from another group. If there are multiple sub-groups that are inherently different from each other, this contributes noise that would hamper a statistical test’s assessment of significant differences. This would also interfere with systems-level analyses like in 2A. If three sub-groups were present within the STIpos that each had their own unique signature of immune cell subsets, they may be equally different from the STIneg and from each other, in which case there would be no statistically significant difference between STIpos and STIneg. Indeed, the STIpos group appears to have more intra-group variance in Figure 2A, which one might expect if there are unique influences on immune cells from the different STI types (Neisseria, Chlamydia, etc.). One recommended way to test for this is ANOSIM. Further exploration of the effects of different STI’s within STIpos is merited. Do any of the sub-groups cluster away from the STIneg? Do they exhibit differences with regards to skewing of immune cell subsets?

3.The authors find that Chlamydia and Neisseria were enriched in the STIpos group consistent with the inclusion criteria for this group. The authors also find five other groups enriched in STIpos (Sneathia, Staphylococcus, Corynebacteriaceae, Bulleidia, and Hemophilus). Calling these taxa “pathogenic” is potentially problematic in that they are commensals that are present in healthy human skin (Staphylococcus, Corynebacteriaceae, and Hemophilus), vagina (Sneathia), and gut (Bulleidia). For example, Staphylococcus genus taxa are present in the skin of nearly every human sampled in large studies (https://www.ncbi.nlm.nih.gov/pmc/articles/PMC4185404/), and though Staphylococcus aureus is a well-known cause of harmful infection, most nares are colonized with it without overt infection or inflammation. “Ectopic” colonization of the gut with taxa from other body surfaces is associated with inflammation and these taxa can themselves stimulate inflammation (https://pubmed.ncbi.nlm.nih.gov/29051379/), so I believe the authors’ observations are of interest and are important, but calling them “pathogenic” may be an over-simplification. The authors may consider using the term of Atarashi et al. of ‘ectopic’ and noting these taxa typically colonize other body surfaces, or otherwise softening the idea that these are overt pathogens.

**Part III – Minor Issues: Editorial and Data Presentation Modifications**

Reviewer #1: 1.The authors should include discussion of recent work in NHPs which suggest that the composition of the microbiome can influence SIV acquisition via low dose intrarectal challenge.

2.The authors should also discuss previous work (32415070) suggesting that the best predictor of HIV associated dysbiosis is nadir CD4 count, arguing that dysbiosis is not common in HIV until very late in the disease. The authors should also discuss previous work demonstrating that the microbiome of MSM is different than that of MSW (27077120)

3.The OTU bar graph in Figure 3 should be arranged by phyla.

4.The use of antibiotics to treat the STIs is mentioned in the manuscript, but their use will obviously influence the composition of the microbiome. This needs to be further fleshed out in the figures and in the table describing the cohort. Which antibiotics, for how long, can the authors break up Abx treated individuals vs untreated in the figures? Do the authors have any longitudinal data (particularly on the microbiome) pre/post Abx use?

5.The authors should provide representative flow cytometry gating strategies in the supplemental figures.

Reviewer #2: 1)Bacterial pathogens that spread by fecal:oral transmission like Campylobacter and Shigella have received attention for spreading sexually among MSM. Were these seen in this cohort? 

2)Some minor typos grammatical: 

a.Page 6 first line of results: “enrolled intro”

b.Page 10 last line: 16s should be 16S (same Page 11 second line and Fig 3 legend)

3)Table 1: not all abbreviations defined in legend (e.g. GC, CT, TP)

4)Would be nice to have more info on how “rectal biopsy and secretion sampling” was performed in main text– by “secretion sampling” are you referring to the rectal swabs for microbiome?

5)Please provide IRB approval number

6)Would be nice to have the gating strategies better described in this paper (supplemental materials if needed for space reasons) rather than just referencing previous publications. Same for the extraction and sequencing methods used to assess the microbiome.

7)Page 9 line 2” confused by the statement “two-dimensional principal components analysis (PCAs) were constructed and analyzed by PERMANOVA” – I may be unfamiliar with this particular application but PERMANOVA is typically run on a distance matrix, not on the results of a PCA analysis. I have seen it coupled with principal coordinates analysis visualizations (PCoA) – because the PCoA might have been drawn using the same distance matrix. If this was done please indicate the distance metric used. If not, please explain in a little more detail here or in the methods how a “two-dimensional PCA” was performed and input into PERMANOVA. Also might want to consider using ADONIS because could evaluate HIV and STI in a single model.

8)Page 10 – for p-values reported – should give more detail on the statistical tests employed. E.g. for alpha diversity measures (Chao 1 etc) – was this a T-test? For beta diversity (Bray-Curtis) was this PERMANOVA? For differential abundance – should consider using a compositionally aware analysis method like ANCOM-BC.

9)Page 10: More details should be given somewhere on how microbiome beta diversity was calculated. Was this done on the ASV table or a taxa summary table and with what software package? Would suggest applying phylogenetically aware metric like unweighted/weighted UniFrac as could add more resolution potentially. 

10)Figure 3 B,C – and associated manuscript text on page 11 – should indicate statistical test used (PERMANOVA) should consider using UniFrac.

11)Bottom page 11 – were these analyses done on genus level taxa and if so defined with which classifier? Methods list 2 different classifiers (Silva and greengenes) and things listed are something at genus and sometimes at family level.

12)How were the 43 individuals selected for RNASeq analyses stratified across the 4 cohorts? (page 12)

13)Page 12 – it is stated that all of the significant genes that differed between YMSM with and without HIV contribute to immune function and tissue homeostasis. How was this determined and in what way? 

14)Last line of page 14 – I thought most prior studies show decrease of MAITs with HIV but you show increase, but says consistent with prior studies

Reviewer #3: Abbreviations for the STIs will be helpful for readers if defined in the Table legend (Table 1) as they are in the text.

PLOS authors have the option to publish the peer review history of their article (what does this mean?). If published, this will include your full peer review and any attached files.

Reviewer #1: No

Reviewer #2: **Yes: **Catherine Lozupone

Reviewer #3: No

Figure Files:

Data Requirements:

Reproducibility:

References:

---

## [Decision Letter · Decision Letter 1]

10 May 2023

Dear Dr Van Doren,

We are pleased to inform you that your manuscript 'HIV, asymptomatic STI, and the rectal mucosal immune environment among young men who have sex with men' has been provisionally accepted for publication in PLOS Pathogens.

Best regards,

Daniel C. Douek

Academic Editor

PLOS Pathogens

Susan Ross

Section Editor

PLOS Pathogens

Kasturi Haldar

Editor-in-Chief

PLOS Pathogens

orcid.org/0000-0001-5065-158X

Michael Malim

Editor-in-Chief

PLOS Pathogens

orcid.org/0000-0002-7699-2064

Reviewer Comments (if any, and for reference):

Reviewer's Responses to Questions

**Part I - Summary**

Reviewer #1: The authors have addressed the concerns raised by all reviewers.

Reviewer #2: My concerns/comments were nicely addressed. I only have one very minor point below.

Reviewer #3: The authors have addressed my comments in their revision, and I believe the study is a valuable contribution to our evolving understanding of what may stimulate ongoing inflammation in HIV. One final comment: It may be useful to show abundances of the potentially pathogenic taxa broken apart into the four different groups (HIV-STI-, HIV+STI-, HIV-STI+, HIV+STI+). I understand power is limited, but it may be interesting for the field to understand hints of whether STI-mediated perturbations in microbiome composition may be more exaggerated in HIV+ individuals.

**Part II – Major Issues: Key Experiments Required for Acceptance**

Reviewer #1: (No Response)

Reviewer #2: None

Reviewer #3: (No Response)

**Part III – Minor Issues: Editorial and Data Presentation Modifications**

Reviewer #1: (No Response)

Reviewer #2: With regard to my comment 11, I still think it is a little unclear in the revised whether done at genus or family level. It appears that all analyses was done at genus level but some were only assigned to family level so in those cases it is stated e.g. that the family differed. However, it is an important distinction that these only represent the bacteria in that family that were not classified at the genus level. bacteria in that family that were classified at the genus level are subtracted out. I usually would indicate this by specifying that all analyses were conducted at genus level and those are e.g. "unclassified Lachnospiraceae"

Reviewer #3: (No Response)

PLOS authors have the option to publish the peer review history of their article (what does this mean?). If published, this will include your full peer review and any attached files.

Reviewer #1: No

Reviewer #2: **Yes: **Catherine Lozupone

Reviewer #3: No

---

## [Editor Report · Acceptance letter]

24 May 2023

Dear Dr Van Doren,

We are delighted to inform you that your manuscript, "HIV, asymptomatic STI, and the rectal mucosal immune environment among young men who have sex with men," has been formally accepted for publication in PLOS Pathogens.

Best regards,

Kasturi Haldar

Editor-in-Chief

PLOS Pathogens

orcid.org/0000-0001-5065-158X

Michael Malim

Editor-in-Chief

PLOS Pathogens

orcid.org/0000-0002-7699-2064